# Akwenda intervention programme for children and youth with cerebral palsy in a low-resource setting in sub-Saharan Africa: protocol for a quasi-randomised controlled study

Gillian Saloojee,[1] Francis Ekwan,[2] Carin Andrews,[3] Diane L Damiano,[4] Angelina Kakooza-Mwesige,[5] Hans Forssberg [ID] [3]

► Prepublication history and additional materials for this paper are available online. To view these files, please visit the journal online (http://dx.doi.org/10.1136/bmjopen-2020-047634).

For numbered affiliations see end of article.

**Correspondence to**
Dr Hans Forssberg;
Hans.Forssberg@ki.se

## ABSTRACT

**Introduction** Cerebral palsy (CP) is the most common childhood-onset motor disorder accompanied by associated impairments, placing a heavy burden on families and health systems. Most children with CP live in low/middle-income countries with little access to rehabilitation services. This study will evaluate the Akwenda CP programme, a multidimensional intervention designed for low-resource settings and aiming at improving: (1) participation, motor function and daily activities for children with CP; (2) quality of life, stress and knowledge for caregivers; and (3) knowledge and attitudes towards children with CP in the communities.

**Methods** This quasi-randomised controlled clinical study will recruit children and youth with CP aged 2–23 years in a rural area of Uganda. Children will be allocated to one of two groups with at least 44 children in each group. Groups will be matched for age, sex and motor impairment. The intervention arm will receive a comprehensive, multidimensional programme over a period of 11 months comprising (1) caregiver-led training workshops, (2) therapist-led practical group sessions, (3) provision of technical assistive devices, (4) goal-directed training and (5) community communication and advocacy. The other group will receive usual care. The outcome of the intervention will be assessed before and after the intervention and will be measured at three levels: (1) child, (2) caregiver and (3) community. Standard analysis methods for randomised controlled trial will be used to compare groups. Retention of effects will be examined at 12-month follow-up.

**Ethics and dissemination** The study has been approved by the Uganda National Council for Science and Technology (SS 5173) and registered in accordance with WHO and ICMJE standards. Written informed consent will be obtained from caregivers. Results will be disseminated among participants and stakeholders through public engagement events, scientific reports and conference presentations.

**Trial registration number** Pan African Clinical Trials Registry (PACTR202011738099314) Pre-results.

## INTRODUCTION

Cerebral palsy (CP) is a life-long motor disorder due to non-progressive disturbances

## Strengths and limitations of this study

► This is the first registered randomised controlled clinical trial of a comprehensive, multidimensional intervention programme for children over the age of 2 years and youth with cerebral palsy in a low-income and middle-income country.

► The core of the programme is based on principles derived from the International Classification of Functioning, Disability and Health (ICF), evidence-based interventions and universal health coverage.

► The study will address several ICF categories and include a battery of objective, validated and adapted outcome measures that will measure a broad spectrum of intervention outcomes at the level of the child, the caregiver and the community.

► The small sample size will make it difficult to identify small effects and it will not be possible to identify which of the components is most effective.

of the developing fetal or infant brain.[1] In addition to gross and fine motor impairments, children with CP often have associated impairments in hearing, vision, cognition and communication, as well as behavioural problems and seizures. Feeding difficulties result in a high prevalence of malnutrition.[2] Apart from access to mainstream healthcare, social services and education, children with CP require services for their special needs. These needs are not met in many low/middle-income countries (LMICs) where children with developmental disabilities often are neglected by primary healthcare and education services and face discrimination and exclusion.[3 4] Discrimination often stems from a lack of knowledge, prejudice and cultural norms, which in turn lead to stigma and entrenched social exclusion.[5–7] LMICs are characterised by a paucity of

health professionals specialised in childhood disability including physiotherapists, occupational, and speech and language therapists,[8–10] as well as a lack of assistive devices.[11] Professional consensus reports from Africa illustrate an immense lack of services for children with neurodevelopmental disorders with limited health worker and caregiver awareness, poor community attitudes and practices, lack of health worker training, unclear treatment or referral guidelines, and a lack of diagnostic facilities or feasible health interventions.[12–14]

The scarcity of rehabilitation services has resulted in families being left alone to care for the child with CP, neither knowing the cause and the life-long prognosis, nor given any advice on how to best care for the child. This situation contributes to the high risk for stress, isolation, depression and reduced quality of life for the caregivers.[15–18] In the absence of health professionals, training of family members has become a strategy to address the workforce gap, for example, the Caregiver Skills Training programmes for children with developmental delays and disabilities.[19] A recent initiative for CP in LMICs is a caregiver training package delivered by health workers (Getting to know cerebral palsy),[20] which aims to empower caregivers and improve care and support for children, recently studied in Ghana.[21 22] A comprehensive parent-delivered intervention programme for early intervention for infants at high risk to develop CP due to adverse events at birth or shortly after, (Learning through Everyday Activities with Parents; LEAP_CP) is currently being evaluated in a randomised controlled trial in Bangladesh.[23] In addition to parent education, LEAP_CP comprises goal-directed motor and cognitive skills training and strategies to enrich the home environment. Another early intervention parent-delivered programme, the 'ABAaNA EIP' is currently being trialled in Uganda for infants under 12 months.[24] It comprises 1 home visit and 10 modules delivered by trained expert parents to groups of 6–10 families of infants once every 1–2 weeks over a 6-month period. There are thus several emerging intervention programmes for at-risk infants and young children with CP in low-resource settings based on evidence derived from high-income countries (HICs). However, as yet, there is no evidence-based programme nor a programme targeting children over 2 years of age in LMICs.

The neglect and lack of services and support for children with CP in LMICs are underscored by limited information about the epidemiology of CP in these countries. In 2015, our research team published the first population-based study on children with CP in sub-Saharan Africa, reporting a 50% higher prevalence than HICs (a crude prevalence of 3.1/1000,[25] vs 1.8–2.3 in HICs.)[26–29] The finding of a higher prevalence of CP has been confirmed in a recent community-based key informant study from Bangladesh[30 31] and is consistent with some previous estimates from different LMIC populations.[32–35] Many children in our study had severe motor impairments often in combination with intellectual disability, impairments in communication and seizures. None had received regular rehabilitation services. Only 8% of those unable

to walk had wheelchairs, and none had assistive devices for walking, hearing, seeing or communication. Only one-third of the children attended school. Notably, the Ugandan cohort of children with CP exhibited significantly lower developmental trajectories of mobility and self-care skills when compared with a similar CP cohort of Swedish children. We concluded that the needs for children with CP and their families in rural Uganda were not met, illustrated by the scarcity of health services, assistive devices and low school attendance, and that the absence of interventions and support contributed to the poorer development of mobility and self-care skills.[4]

To address this unsettling situation, we have developed a comprehensive and holistic intervention programme for children with CP living in rural, low-resource settings, which we plan to implement and evaluate in this cohort of children with CP in rural Uganda. This Akwenda CP programme was developed by an international team of experts on CP and rehabilitation, an academic team from Uganda, a team from Malamulele Onward in South Africa, and a local team of therapists and social workers living in the region in a three-step process described in online supplemental file 1. The first step involved identifying key principles based on current concepts of disability, rehabilitation and interventions for children with CP. In the second step, the team set up three general programme goals based on the principles and needs of children with CP and their families, which had been identified in previous studies.[2 4 25 36] In the third step, the team developed the Akwenda CP programme that comprises five components, based on the principles and aimed at achieving the goals formulated in the previous steps (see online supplemental figure 1 and online supplemental tables 1 and 2).

## Aims and hypotheses

This protocol describes the method and implementation of the Akwenda CP programme study, which aims to evaluate the programme's feasibility and impact on children, caregivers and society. The specific aims are to:

1. Determine the efficacy of the intervention on the motor function, daily activities and participation of children with CP.
   Hypothesis 1: children with CP receiving the intervention will improve their motor function, enhance their activity and increase their level of participation compared with children receiving usual care.
2. Describe the impact of the intervention on the behaviour and mental health of caregivers of children with CP.
   Hypothesis 2: caregivers of children with CP receiving intervention will have a better quality of life, reduced levels of anxiety and stress, and increased levels of knowledge and practical caregiver competence compared with caregivers receiving usual care.
3. Determine the impact of the intervention on the community awareness of childhood disability and attitudes towards children with disability and their families.

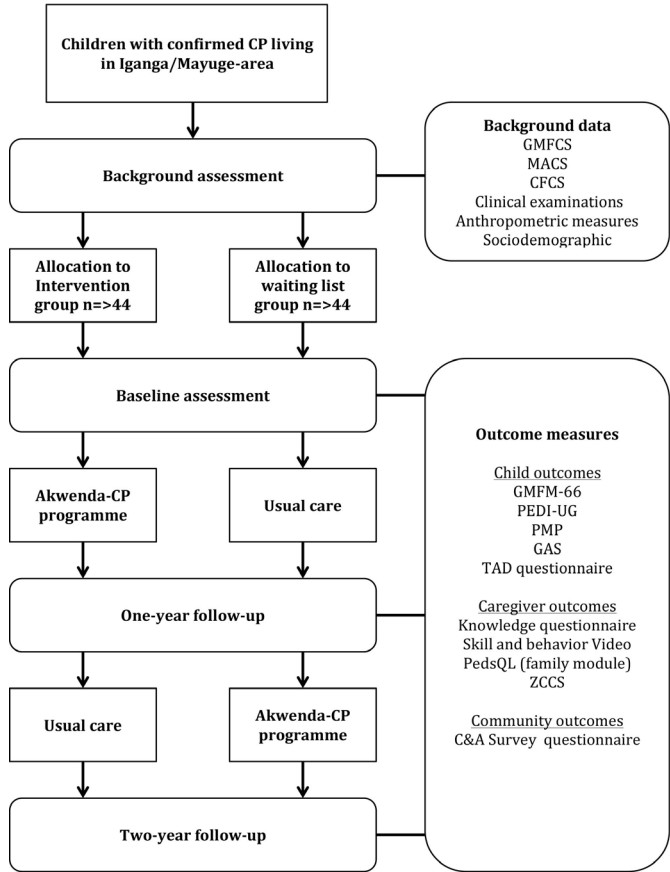

**Figure 1** Flow chart for children in the Akwenda Cerebral Palsy (CP) Intervention Study. C&A, communication and advocacy; CFCS, Communication Function Classification System; GAS, Goal Attainment Scale; GMFCS, Gross Motor Function Classification System; GMFM-66, Gross Motor Function Measure; MACS, Manual Ability Classification System; PEDI-UG, Uganda version of Pediatric Evaluation of Disability Inventory; PedsQL, Pediatric Quality of Life Inventory; PMP, Picture my Participation; TAD, technical assistive device; ZCCS, Zimbabwe Caregiver Challenges Scale.

Hypothesis 3: stakeholders, teachers, health professionals and the lay community will have increased their knowledge about childhood disability and the barriers limiting their participation in society. Their perception of disability will become more positive.

## METHODS
### Study design
The project is designed as a quasi-randomised controlled clinical study conducted according to the Standard Protocol Items: Recommendations for Interventional Trials guidelines (see figure 1). Children with confirmed diagnoses of CP will be assigned to either a study group or control group, based on geographical regions. The study group will receive the Akwenda CP programme during the first year, while the control group will be on a waitlist and receive usual care. The control group will receive the intervention after 1 year. Children and caregivers of the first intervention group will be encouraged to continue with the parent-led workshops and to implement new caregiving skills and behaviours during the second year. Both groups will be assessed prior to the start of the intervention, after 1 year and after 2 years.

### Study site
The study will be performed at the Iganga/Mayuge Health Demographic Surveillance Site (IM-HDSS), which was set up in collaboration between Makerere University and Karolinska Institutet. It comprises 65 villages including 15 964 households, and >85 000 individuals of whom approximately 50% are under 18 years. The populations are predominantly rural subsistence farmers living under the poverty level, but 20% of the population resides in the semiurban Iganga town. The HDSS includes 1 general hospital (Iganga Hospital), 27 health centres and 132 private health providers.

### Participants and recruitment
Participants will be from a population-based cohort of children with CP identified from previous screening carried out at IM-HDSS in 2015 using the definition of CP Surveillance in Europe.[25 37] In addition, children with CP aged 2–6 years living in the same area will be recruited by the community development officer. The CP diagnosis will be confirmed by a child neurologist using the same definitions. Hence, the study population will be between 2 and 23 years at the onset of the intervention study.

### Randomisation and blinding
Since the community and advocacy intervention will target the community, all children from a particular village need to be in the same group. Children will thus be allocated to groups depending on the village where they reside. Neighbouring villages will be clustered together to reduce contamination between groups resulting in two geographically defined groups, which will be matched for age, sex and motor function. The decision regarding which group will receive the intervention first will be decided by a coin flip. A team of therapists and a social worker will administer the assessments and surveys before the study to capture background and baseline data. A group of independent therapists, not involved in the intervention, will administer the outcome measures after the interventions. These examiners will be blinded to group allocation.

### Intervention arm
The Akwenda CP programme comprises five elements (see figure 2, online supplemental tables 1 and 2 for detailed description):

#### Caregiver-led training workshops
The core is a series of seven workshops for caregivers of children with CP. This workshop series is based on the Malamulele Onward Carer-2-Carer Training Programme (MOC2CTP).[38 39] Four caregivers of children with CP from Iganga have been identified and trained to lead the

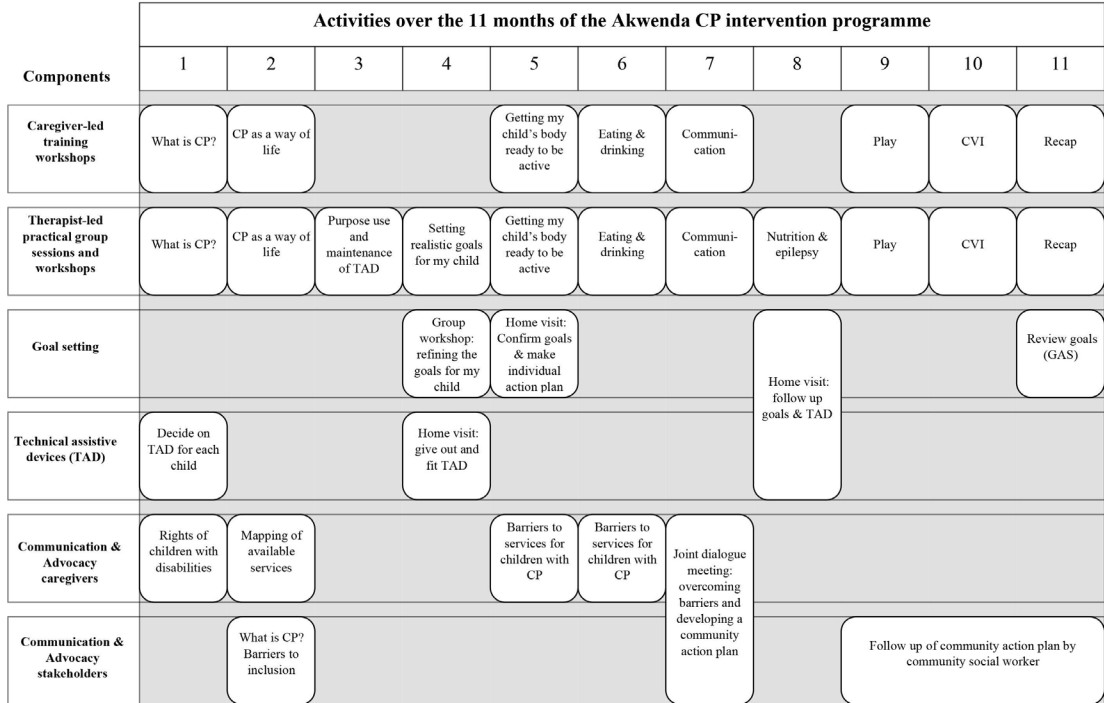

**Figure 2** Timetable for delivering the intervention through the five components over a period of 11 months. CP, cerebral palsy; GAS, Goal Attainment Scale; CVI, Cerebral Visual Impairment; TAD, Technical Assistive Device.

workshops following the detailed MOC2CTP manual (see figure 2 and online supplemental table 1).

### Therapist-led practical group sessions and workshops

Trained paediatric occupational and physiotherapists will run monthly practical sessions for groups of caregivers, based on the material covered in the caregiver-led workshops. The emphasis of the practical sessions is on active caregiver participation and includes training caregivers to facilitate their children's active participation and engagement. The sessions will be structured such that the therapist will be able to give each child individual attention during group activities. The parent facilitators will assist the therapists during these practical sessions. The therapists will in addition run four workshops including (1) technical assistive devices (TADs); (2) goal-based training; (3) nutrition and (4) epilepsy (see figure 2 and online supplemental table 2). To ensure consistency and quality, a therapist manual was developed for the intervention.

### Provision of TADs

TADs are an important part of evidence-based rehabilitation, promoting greater independence and enabling children to perform tasks they were formerly unable to accomplish within mobility, self-care, eating and communication domains.[40] Each child's need for TAD will be assessed by therapists at baseline, and the children and caregivers will be involved in deciding what TAD they need. Examples are wheelchairs, standing frames, posture support chairs, walking devices, parallel bars, communication boards and cut-out feeding cups. When possible, TAD will be locally manufactured with affordable local

materials, and caregivers will be involved in cost-sharing to ensure affordability and sustainability.

### Goal-directed training

Goal-directed training is a child-active, motor learning approach.[41] Several studies have demonstrated successful goal-directed programmes in HICs,[42–45] while there are no reports from LMICs. The child and caregiver, supported by a therapist, will identify three realistic goals and create action plans for how to achieve them through daily training and practise in the home environment. The goal-setting procedure will start during the fourth month. By this time, the therapist has become acquainted with the child, and the caregiver has attended the caregiver-led workshops and hence will have a more realistic understanding of the child's potential. Children-issued TAD will have a fourth goal relating to their use.

### Communication and advocacy for behavioural and social change

The communication and advocacy (C&A) component will address stigma, discrimination and exclusion of children with disabilities, and is based on Communication for Development, a systematic and evidence-based strategy to promote positive social and behavioural changes.[46] The broad themes for C&A are human rights of children with disabilities and barriers to services and how to overcome them. C&A will be run by a community social worker using community meetings and radio talk shows. Community meetings will involve political and religious leaders, teachers and health workers. Caregivers and community stakeholders will participate in a joint meeting to develop

a community action plan for how to make services more inclusive.

## Standard care arm

Standard or usual care refers to that available for children with CP in the study area. Currently, children have access to primary healthcare and the local hospital. Support and services are rare in this rural setting with no specialised services at the local public hospitals or health centres. There are some private actors and non-governmental organisations working in the surrounding area, which may provide occasional sessions of physiotherapy or occupational therapy, and unpredictable donations of TADs such as wheelchairs or crutches.[4] Caregivers in the usual care group will be asked to keep a monthly care diary during the waiting period.

## Sample size

From the initial cohort of 97 children with CP, 82 children had survived.[36] In order to include younger children, we have identified 20 additional children with CP living in the area. From these two samples, 88 children will be recruited to form two comparable groups each with 44 children. In addition, the primary caregiver of each child will be included in the two arms, and 40 community representatives will be included in the intervention arm for evaluation of the C&A component.

To explore whether this sample size was sufficient for detecting a true effect of the intervention, we did power calculations to determine what mean difference in Pediatric Quality of Life Inventory (Peds-QL; primary outcome measure for caregivers) scores was needed between arms. We used a Peds-QL mean of 12.5 and SD of 18.6, from a previous study in Ghana on caregivers of children with CP.[22] In our population with 44 per group allowing for a 10% non-participation at follow-up, a mean difference of at least 12.5 points between groups is needed for 80% power and a type I error of 0.05. Considering that the mean improvement after a parent training intervention in Ghana in Peds-QL was +31.3 (95% CI 25.6 to 37.0), this seems plausible.[22]

## Measures

To evaluate the multidimensional study aims, a series of outcome measures will be used before (baseline) and 1 and 2 years after the intervention at the following levels: (1) child, (2) caregiver and (3) community (see figure 3). The outcome measures were selected and modified by the international team of experts, a team from Malamulele Onward in South Africa, and a local team of therapists and social workers living in the region. We selected gold standard outcome measures when possible, but since the context in LMICs differs from HICs, some tools needed to be modified, and others were selected among tools previously used in other low-resource settings (see online supplemental file 1).

## Background characteristics

Background data will be collected on all children before allocation to groups. This includes sociodemographic information and medical history: Gross Motor Function Classification System, Manual Ability Classification System, Communication Function Classification System; clinical and nutritional examinations including anthropometric data; information on school attendance, care seeking, and need for and current usage of TAD.

## Child-related outcome measures
### Gross Motor Function Measure-66

The Gross Motor Function Measure-66 will be used as the primary child-related outcome. It is a clinical tool regarded as the gold standard when evaluating change in gross motor function in children with CP.[47]

### Uganda version of Pediatric Evaluation of Disability Inventory

This is a parent-reported measure of the child's independence in self-care, mobility and social function widely used in both research and clinical practice. The Pediatric Evaluation of Disability Inventory was developed in North America and has recently been culturally adapted and translated to the Ugandan context.[48 49]

### Picture my Participation

Picture my Participation is an instrument designed to assess the participation of children, particularly in LMICs.[50] It measures attendance and involvement in 20 activities.

### Goal Attainment Scale

Goal Attainment Scale (GAS) is an objective method of quantifying goal attainment that has been widely applied to children with CP.[41] Goals are scored on a 5-point Likert-type scale from no change at all to attainment of goal at much more than the expected level. This measure will only be used in the intervention group.

### Evaluation of TAD

A goal for the use of the assistive devices will be set for each child in the intervention group, and the GAS scale will be used to evaluate the use of assistive devices. User satisfaction will be evaluated at the end of the intervention using a questionnaire based on the Quebec User Evaluation of Satisfaction with Assistive Technology adapted to Uganda.[51]

## Caregiver-related outcome measures
### A Knowledge Questionnaire evaluating caregiver's learning outcomes

A Knowledge Questionnaire evaluating caregiver's learning outcomes from the caregiver-led workshops will be administered before and after the workshop. This questionnaire comprises 26 statements, with three possible responses: agree, disagree or not sure. It is based on the Knowledge Questionnaire developed to evaluate the MOC2CTP. It was modified during the piloting phase; two statements were removed because they were

# Akwenda CP programme

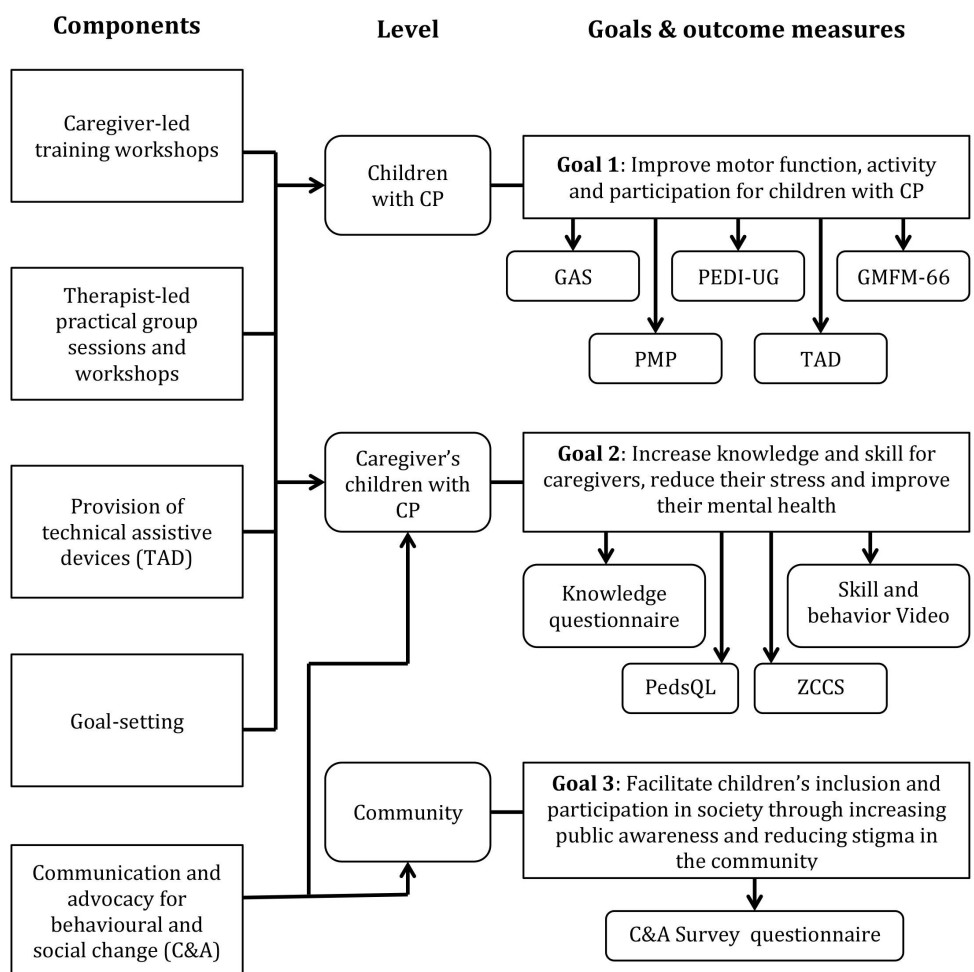

**Figure 3** Akwenda CP programme overview with the five components, three levels, three goals and all outcome measures. See online supplemental file 1 for detailed background and design of the programme. CP, cerebral palsy; GAS, Goal Attainment Scale; GMFM-66, Gross Motor Function Measure; PEDI-UG, Uganda version of Pediatric Evaluation of Disability Inventory; PedsQL, Pediatric Quality of Life Inventory; PMP, Picture my Participation; ZCCS, Zimbabwe Caregiver Challenges Scale.

ambiguous and one statement was rewritten to provide greater clarity.

### Caregivers' skill and behaviours

Interaction, skills and behaviours of caregivers performing two everyday activities with their child: that is, dressing and undressing; and feeding in the child's home will be captured in two videos, each 4 min long. The assessment of the video clips has been developed and used by Malamulele Onward and further adapted by the research team for this study. Five observable behaviours will be scored using a 4-point Likert scale, with 0 being 'none of the time' and 3 being '90% or more of the time'.

### Caregivers' quality of life

Caregiver quality of life will be assessed using the 36-item PedsQL Family Impact Module, which is a validated and reliable measure.[52] This was successfully used in a similar study in Ghana,[21] and has been translated into Luganda, the local language.

### Caregivers' stress

Caregiver stress will be assessed using the Zimbabwe Caregiver Challenges Scale.[53] This tool was recently developed for caregivers of children with lifelong disabilities living in low-resource African settings.

### Community-related outcome measures
### Communication and Advocacy Survey Questionnaire

This tool is based on the Christian Blind Mission Inclusive Development Tool Kit[54] and comprises 44 items measuring the knowledge, attitudes and practices of community members. The tool, available in both English and Lusoga, has been developed by the research team and pretested on community members in Uganda.

### Data collection, management and analysis
### Organisation and management

The Akwenda CP programme is a study within the CURIE research consortium, a collaboration between the Makerere University College of Health Sciences,

Kampala, Uganda and Karolinska Institutet, Stockholm, Sweden, and clinical researchers from South Africa and the USA. The project will be guided by a steering group ensuring the conduct and coordination of the study. There will be a local Site Steering Committee consisting of the principal investigator (PI) (AK-M), an administrative project manager, a PhD student and a parent carer. The project manager will be coordinating the study on site and report to the PI.

## Data collection and monitoring

Two therapists not involved in the intervention will administer the baseline and outcome assessments. They have preknowledge of the project from participating in the development of the intervention, and will be trained over a period of 4 weeks in the assessments. The PhD student, also a therapist, will assist in the assessments and be responsible for data collection, storage and analysis. A data manager will supervise the overall organisation, storage and security of the data and ensure information is recorded accurately.

## Data management and analysis

Data will be collected in Open Data Kit using tablets, and electronic data will be managed through a secure database. The use of electronic data collection with built-in quality control will minimise missing or inaccurately entered data. The researchers will supervise data collection and monitor the database in real time to address any quality issues that might be arising. Data will be analysed using Stata version 14.2. Standard analysis methods for a randomised controlled trial, including general linear mixed models, will be used to make comparisons between the intervention group and the waitlist group on an intention-to-treat basis. Effect estimate will be presented as a mean difference and 95% CI. Secondary outcome analyses will use similar methods to compare all other outcomes between groups immediately post-intervention and again 1 year later. The GAS goal attainment scores for the intervention group will be analysed with paired t-tests on a within-group basis. A senior statistician will guide and assist with the analysis.

## Patient and public involvement

The Akwenda CP programme and the study design have been developed in collaboration with caregivers, community leaders and local health professionals. All of these stakeholders were invited to a dissemination event in January 2019 where the results from the first screening study from 2015 were reported. At the same time, plans for a 4-year follow-up study and this intervention study were presented and discussed. Local therapists and social workers from the region, together with four caregivers, have thereafter participated in the development and adaptation of the intervention programme (see online supplemental file 1), and in designing and planning the study. The parent-led workshop series, a core part of the intervention programme, was developed and refined by a team of therapists and the parents of children with CP in South Africa.

## Ethics and dissemination
### Ethics

The research project has been reviewed by the Higher Degrees Research and Ethics Committee, School of Public Health, College of Health Sciences, Makerere University (Ref: 727) and approved by the Uganda National Council for Science and Technology (SS 5173). Written informed consent for study participation will be obtained from the mother or legal guardian, and, if possible, assent will be obtained from children above 8 years. The electronic data will be managed through a secure database. Participants' identities will not be revealed in any publications or reports emanating from this study. Video file names will be de-identified. A study protocol including all items of the WHO Trial Registration Data Set has been completed and the study has been registered in accordance with WHO and ICMJE standards. Any amendment of the study protocol requires approval from the Research and Ethics Committee.

### Safety

As the study is not a drug trial, there is no Data Safety Monitoring Committee. A COVID-19 risk management protocol has been developed and approved by the Higher Degrees Research and Ethics Committee, School of Public Health, College of Health Sciences, Makerere University and will be strictly adhered to. Any adverse events associated with the study will be screened weekly by the treating therapists, and reported to the PI, and referred to the Ethics Committee if considered serious. Any decision to describe interim results or terminate the trial will be made by the investigators with oversight from the Ethics Committee.

### Dissemination

Locally within Iganga-Mayuge, a stakeholders' dissemination event will be organised. Families involved in the study along with the key players in the health, community, policing, media and disability groups will be invited to participate.

At the national level, findings will be disseminated through the Ministry of Health, non-communicable diseases platform, and collaboration with health regulatory councils and professional associations will result in professional development courses that use the study findings to be disseminated to nearly all the health practitioners and health policy planners/implementers working in Uganda. Findings will also be disseminated through print media that will enable non-health workers to access information regarding the study findings.

For the international community, study findings will be published through open access, peer-reviewed scientific journals and presented at international conferences. Several authors are in the International Alliance of Academies of Childhood Disability (IAACD), and these findings

will be important when developing recommendations for good clinical practice and will be disseminated through lectures and webinars organised by the IAACD.

## DISCUSSION

To our knowledge, this is the first intervention programme for children and youth with CP in an LMIC that provides a holistic approach with a broad spectrum of interventions aimed at children and their caregivers as well as the wider community. The intervention is designed to be affordable in a low-resource setting where many families live at or below the poverty level, and with a scarcity of therapists and other rehabilitation professionals. The core of the intervention is a carer-to-carer training programme. Intervention programmes in Ghana and West Bengal have previously used parent training programmes,[21 23] however, that training was either delivered by health workers or community disability workers, only some of whom were mothers of children with disabilities.

Currently, few evidence-based interventions have been evaluated in LMICs. The activities of the Akwenda CP programme are based on best clinical practice derived from clinical studies in HICs, and adapted to the conditions in this rural part of Uganda in accordance with the WHO's goal of universal health coverage and universal access.[55] The multiple activities add to the complexity of the intervention as well as the evaluation. It will therefore not be possible to study the effects of individual components. Our aim is rather to study the impact of the entire intervention as well as the programme's feasibility, cost implications, effectiveness, accessibility and acceptability.

Most intervention studies focus on children under the age of 12 years.[21–24] Our study includes a much wider age range, up to the age of 23 years. The majority of children and youth have not had access to early intervention or to regular therapy, which is the situation in many LMICs. By including a wider age range of participants, findings from this study will be relevant to all age groups, not just younger children. Information from this study will also shed light on the extent to which positive change is still possible in youth with CP who have received very little intervention during early life.

**Author affiliations**
[1]Independent Researcher, Johannesburg, South Africa
[2]Department of Occupational Therapy, Mulago National Referral Hospital, Kampala, Uganda
[3]Department of Women's and Children's Health, Karolinska Institutet, Stockholm, Sweden
[4]Department of Rehabilitation Medicine, The National Institutes of Health Clinical Center, Bethesda, Maryland, USA
[5]Department of Pediatrics and Child Health, Mulago Hospital/Makerere University, Kampala, Uganda

**Collaborators** The development of the intervention was done in collaboration with a team of four local therapists from the region: Sauba Kamusiime OT, Catherine Ntanya OT, Edward Opio OT and Eric Kintu PT; one social worker: Godfrey Wanjala from Iganga; four caregivers: Babra Esaete, Margret Nalweiso, Rose Naigaga and Robinah Nakaziba who were trained to become parent facilitators by Lydia Ngwana, a master trainer from Malamulele Onward in Johannesburg, South Africa. The parent facilitators and master trainer were also involved in adapting the carer-to-carer programme to the local conditions in rural Uganda. Keron Ssebyala was involved in the project administration and coordination, and also supported the practical work in planning the intervention programme and study, and the training of the staff. Tom Babalanda was involved in coordinating logistical requirements during field activities including training. IM-HDSS supported with data management within their system at the Iganga field site.

**Contributors** GS led the development of the intervention programme based on her previous experience at Malamulele Onward. She was also involved in designing the study protocol, drafted the manuscript and approved the final manuscript as submitted. FE was involved in leading the intervention programme development, in particular the C&A component, and taking part in the design of the study protocol, and drafted the manuscript and approved the final manuscript as submitted. CA has been involved in programme development and design of the study protocol and has revised drafts of the manuscript and approved the final manuscript as submitted. DD has been involved in programme development and design of the study protocol and has revised drafts of the manuscript and approved the final manuscript as submitted. AK-M conceptualised the study and was involved in programme development and design of the study protocol and revised drafts of the manuscript, and approved the final manuscript as submitted. HF conceptualised the study, secured funding for the study, was involved in programme development, led the design of the study protocol, drafted the manuscript, approved the final manuscript and was responsible for submission.

**Funding** We acknowledge the funding of the project by the Swedish Research Council (2017-05474) and by the Foundation Frimurare Barnhuset in Stockholm.

**Disclaimer** The funders had no role in study design; collection, management, analysis and interpretation of data; writing of the report and the decision to submit the report for publication.

**Competing interests** None declared.

**Patient consent for publication** Not required.

**Provenance and peer review** Not commissioned; externally peer reviewed.

**ORCID iD**
Hans Forssberg http://orcid.org/0000-0003-2069-6618

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
