## [Reviewer comments · BMJ Open]

ARTICLE DETAILS

TITLE (PROVISIONAL)	Akwenda intervention programme for children and youth with cerebral palsy in a low-resource setting in sub-Saharan Africa: protocol for a quasi-randomised controlled study
AUTHORS	Saloojee, Gillian; Ekwan, Francis; Andrews, Carin; Damiano, Diane; Kakooza, Angelina; Forssberg, Hans

VERSION 1 – REVIEW

REVIEWER	Stephanie Tremblay McGill University, Canada
REVIEW RETURNED	06-Jan-2021

GENERAL COMMENTS	Review for BMJ Open January 2021 Article: Community Based Intervention Programme for Children with Cerebral Palsy in a Low-Resource Setting in SubSaharan Africa: protocol for a quasi-randomized controlled study bmjopen-2020-047634 Authors: Saloojee, Gillian; Ekwan, Francis; Andrews, Carin; Damiano, Diane; Kakooza, Angelina; Forssberg, Hans This is a very important study that has the potential to make a big difference in the targeted population. The study is overall well-described, original and scientifically sound. It is a wonderful example of collaboration between different countries and institutions. There are a few points that could be further clarified. It would be helpful to have a bit more information about usual care looks like in this setting, for example if children receive follow up once a year at the hospital (or at what frequency, if any of them have access to rehabilitation, etc.). Sometimes it says it will be 11 months for the intervention and sometimes 1 year, please check the manuscript for consistency throughout. For the measures, please consider the burden of administering all these questionnaires and assessments (justify the need). Also for the measures, specify if they are gold standards, or if there were other tools that could have been used instead. For the 3rd objective, it is a big statement to say there was a community shift in attitudes, as there is a limit to the reach of dissemination strategies. Also, the sample size can include the targeted number of community members that will complete the questionnaire to determine community impact of the intervention. Finally for data analysis, it could be specified what will be done
--

	with missing information in the questionnaires or participant drop-outs. Spelling and phrasing to be adjusted: P. 6 of 44 Line 11, mainstream Line 31, advice on how to Line 36, LMICs (be consistent with the s when acronym is used unless referring to one low or middle income country), also line 53, etc. Line 40, rephrase infants at high risk to develop CP as it happens at birth or shortly after and is non-progressive Line 44, skills training Line 46, add a short phrase about the ABAaNA EIP program as was done for LEAP_CP above Line 50, HICs (add s), also line 57, 58 etc. P. 7 Line 40, programme`s Lines 48 and 57, compared to P. 8 Line 25, caregivers Line 44, spell out HDSS acronym P. 9 Line 14, a team of therapists and social workers Line 18, blinded instead of masked Line 45, spell out acronym for TAD (even though it is a few lines below) P. 10 Line 41, spell out acronym for NGO P. 11 Line 9, missing period at the end of sentence. Line 47, capitalize Assistive Line 55, possible responses: agree, disagree or not sure (remove viz.) Line 57, specify main modifications P. 13 Line 43, obtained instead of required Line 44, Participants` P. 14 Line 12, and disability groups, or communities Line 24, and presented at international conferences P. 15 Line 19, missing period at the end of sentence. P. 23 Line 53, support Line 55, ICF`s P. 28 Line 7, remove viz. or clarify
--	--

	Line 19, enable Line 43, walking assistive devices and parallel bars, all of which
--	---

REVIEWER	Melissa Gladstone University of Liverpool
REVIEW RETURNED	23-Jan-2021

GENERAL COMMENTS	Community Based Intervention Programme for children with Cerebral Palsy in Uganda This paper is a protocol for a study on an intervention for children with cerebral palsy for 2 to 23 years in Uganda. It is exciting to see this paper and this paper will add something to the literature by providing so much detail on the measures to be used for outcomes as well as the overall structure of the programme they are planning to provide. I have some few comments only. Introduction: Intro paragraph 2, the authors describe how “the scarcity of rehabilitation services has resulted in families being left alone....”. I wonder if the authors need to describe it as “rehabilitation services” which are the lack and could just stick to “services” as it is not necessarily “rehabilitation” that is the issue, but more support and services in many different ways. Caregiver skills training NOT Care skills training The authors describe other programmes e.g. LEAP-CP or ABANA but they are not clear enough to the reader why they have created a separate programme and what is different about Akwenda in comparison to LEAP-CP or ABANA. I think that one factor is that the children being targeted in Akwenda are older and that the programme is hitting at a number of different levels; child, family, community that may be different to LEAP-CP or ABANA – but could this be a little clearer for the reader? I wondered why they did not use or link with the ABANA programme, which is fine – but what is different about Malamulo Onward parent groups in comparison to ABANA – this could be much clearer. Also – this relates to the “Strengths and Limitations of the study” where the authors describe that they are the “first registered randomised controlled trial....” But in the introduction they describe these other randomised controlled trials. Just more clarification needed. This also links to the discussion where this is mentioned to some extent, but I wonder if it might be useful to be more “up front” and clearer in the introduction. Strengths and Limitations: The authors describe how a strength of the study is that the “core of the programme is based on principals of the ICF”. This is great and novel. The authors then describe how they will use “parent led workshops to account for the scarcity of rehabilitation”.. This is not necessarily novel so I wonder if the authors should concentrate more on the fact that they are using principals of the ICF for the core of their programme and that it is focussed at three different levels; child, family and community – this is more novel. Methods: The authors describe how they will assign the study group and the control group dependent on regions that the children live in. This does not seem to be a “randomised” method of sampling and I wonder if the authors could be clearer as to how this is randomised and exactly what they did in sampling for
--

	randomisation. Actually I can see later that it is discussed in a next section where the authors describe “doing a coin flip”. Recruitment: Children from a previous study with CP are to be recruited and then some new children. How will a diagnosis of CP be confirmed in these new children and could the team mention how it was confirmed in the previous study? Outcomes There is a good range of outcome measures which are being utilised and it is great to see this list of measures. This will be incredibly helpful for other researchers. Should the team mention their primary outcome more clearly (as mentioned in the sample size calculation). Steering committee: I wondered if the team would consider having a parent carer on the steering committee? This might add weight to the study. Please could the authors go over the grammar – it may just be a matter of English but there are places where things are not reading quite right e.g. Page 3 “Strengths and Limitations”; “will be parent-led training workshops which account for the scarcity...” rather than “which accounts for” .
--	--

VERSION 1 – AUTHOR RESPONSE

Reviewer Reports:

Reviewer: 1

Dr. Stephanie Tremblay, McGill University

Comments to the Author:

Very interesting study! This is a very important study that has the potential to make a big difference in the targeted population. The study is overall well-described, original and scientifically sound. It is a wonderful example of collaboration between different countries and institutions.

There are a few points that could be further clarified. It would be helpful to have a bit more information about usual care looks like in this setting, for example if children receive follow up once a year at the hospital (or at what frequency, if any of them have access to rehabilitation, etc.).

In the rural settings it is rare that children receive rehabilitative care, however the few that do may get sessions of general physiotherapy or occupational therapy. Speech and language therapy is only offered at the referral hospital in Kampala. Sometimes if donations of wheelchairs or some other technical assistive devices are available they may also be utilised by these children. The follow up period depends on several factors such as ability of the caregiver to transport the child regularly to the health centre, availability of the required services and therapists, and financial constraints amongst others. In general, the follow up period may be after 1-3 months.

We have added some of this information in the section Standard care arm

Sometimes it says it will be 11 months for the intervention and sometimes 1 year, please check the manuscript for consistency throughout.

The duration of the intervention programme is 11 months, while the study, including pre-and post-intervention assessment will be a year. We have carefully gone through the text and checked for consistency.

For the measures, please consider the burden of administering all these questionnaires and assessments (justify the need). Also for the measures, specify if they are gold standards, or if there were other tools that could have been used instead.

We are using a wide range of assessment tools in this study, since the intervention is complex and targets three different groups. That means, that each target group will not be subjected to all assessments. We have added a figure to illustrate the outcome measures at respective i) child, ii) caregiver and iii) community levels - see figure 3.

We have also added some information on the selection of outcome measures in the main text, and information on gold standard, previous use and development/adaptation for LMIC.

For the 3rd objective, it is a big statement to say there was a community shift in attitudes, as there is a limit to the reach of dissemination strategies.

We agree there are few evidence based dissemination strategies, yet it is crucial to address the stigma if the overall intervention shall succeed. In our communication and advocacy component, we will utilize, active, multifaceted approaches rather than relying solely on passive information transfer which is relatively ineffective. Interventions that incorporate two or more distinct strategies (i.e., that are multifaceted) are more likely to work than single interventions and as such we envisage realizing a change in the attitudes of involved community members, however modest they may be. We do think the presented aim 3 is reachable, but of course, they may not be reached.

3. Determine the impact of the intervention on the community awareness of childhood disability and attitudes towards children with disability and their families.

Hypothesis 3: Stakeholders, teachers, health professionals and the lay community will have increased their knowledge about childhood disability and the barriers limiting their participation in society. Their perception of disability will become more positive.

Also, the sample size can include the targeted number of community members that will complete the questionnaire to determine community impact of the intervention.

We have added primary caregivers and community members in the section on sample size.

Finally, for data analysis, it could be specified what will be done with missing information in the questionnaires or participant drop-outs.

We have adjusted for a 10% drop-out rate in the sample size calculation. We have also built in quality control system in the Open-data kit for data collection, which will prevent missing information.

We have added some information on this in the section on Data management and analysis.

Spelling and phrasing to be adjusted:

Thanks for this help. We have followed all the spelling and phrasing suggestions provided below.

P. 6 of 44

- Line 11, mainstream
- Line 31, advice on how to
- Line 36, LMICs (be consistent with the s when acronym is used unless referring to one low or middle income country), also line 53, etc.
- Line 40, rephrase infants at high risk to develop CP as it happens at birth or shortly after and is non-progressive
- Line 44, skills training
- Line 46, add a short phrase about the ABAaNA EIP program as was done for LEAP_CP above
- Line 50, HICs (add s), also line 57, 58 etc.

P. 7

- Line 40, programme`s
- Lines 48 and 57, compared to

P. 8

- Line 25, caregivers
 - Line 44, spell out HDSS acronym
- P. 9
- Line 14, a team of therapists and social workers
 - Line 18, blinded instead of masked
 - Line 45, spell out acronym for TAD (even though it is a few lines below)
- P. 10
- Line 41, spell out acronym for NGO
- P. 11
- Line 9, missing period at the end of sentence.
 - Line 47, capitalize Assistive
 - Line 55, possible responses: agree, disagree or not sure (remove viz.)
 - Line 57, specify main modifications
- P. 13
- Line 43, obtained instead of required
 - Line 44, Participants`
- P. 14
- Line 12, and disability groups, or communities
 - Line 24, and presented at international conferences
- P. 15
- Line 19, missing period at the end of sentence.
- P. 23
- Line 53, support
 - Line 55, ICF`s
- P. 28
- Line 7, remove viz. or clarify
 - Line 19, enable
 - Line 43, walking assistive devices and parallel bars, all of which

Reviewer: 2

Dr. Melissa Gladstone, University of Liverpool Institute of Translational Medicine

Comments to the Author:

Community Based Intervention Programme for children with Cerebral Palsy in Uganda This paper is a protocol for a study on an intervention for children with cerebral palsy for 2 to 23 years in Uganda. It is exciting to see this paper and this paper will add something to the literature by providing so much detail on the measures to be used for outcomes as well as the overall structure of the programme they are planning to provide.

I have some few comments only.

Introduction: Intro paragraph 2, the authors describe how “the scarcity of rehabilitation services has resulted in families being left alone....”. I wonder if the authors need to describe it as “rehabilitation services” which are the lack and could just stick to “services” as it is not necessarily “rehabilitation” that is the issue, but more support and services in many different ways.

We agree that the term rehabilitation is not optimal when we are dealing with children with developmental disabilities, and in many countries, like in Sweden, we use the term “habilitation”. We have now deleted rehabilitation in “rehabilitation services”, and changed to “health professionals specialized in childhood disability”.

Caregiver skills training NOT Care skills training.
Changed accordingly

The authors describe other programmes e.g. LEAP-CP or ABANA but they are not clear enough to the reader why they have created a separate programme and what is different about Akwenda in comparison to LEAP-CP or ABANA. I think that one factor is that the children being targeted in Akwenda are older and that the programme is hitting at a number of different levels; child, family, community that may be different to LEAP-CP or ABANA – but could this be a little clearer for the reader?

Yes, this is true. Both the LEAP-CP and ABAaNA programmes are early intervention programmes designed for young infants and children under the age of two. Both also target primarily one level – that of parent education and coaching.

To make this clearer to the reader, the words “nor a programme targeting children with CP over two years of age” has been added to the second paragraph of the Introduction in the main article and an additional sentence has been added to the Introductory paragraph of the Supplement.

I wondered why they did not use or link with the ABANA programme, which is fine – but what is different about Malamulo Onward parent groups in comparison to ABANA – this could be much clearer.

An additional sentence has been added to the paragraph Component 1: Caregiver-led training workshops, to explain the differences between the Malamulele Onward parent training programme and the ABAaNA programmes.

Also – this relates to the “Strengths and Limitations of the study” where the authors describe that they are the “first registered randomised controlled trial...” But in the introduction they describe these other randomised controlled trials. Just more clarification needed.

We have clarified the first point in the “Strengths and limitations” section by adding the words “over the age of two years”.

This also links to the discussion where this is mentioned to some extent, but I wonder if it might be useful to be more “up front” and clearer in the introduction.

We have now tried to emphasize that Akwenda is aimed at older children and not an early intervention programme, and we have also emphasized that the intervention is holistic (multi-dimensional) and not only including parent-training.

Strengths and Limitations:

The authors describe how a strength of the study is that the “core of the programme is based on principals of the ICF”. This is great and novel. The authors then describe how they will use “parent led workshops to account for the scarcity of rehabilitation”. This is not necessarily novel so I wonder if the authors should concentrate more on the fact that they are using principals of the ICF for the core of their programme and that it is focussed at three different levels; child, family and community – this is more novel.

We have deleted the parent-led workshops, which indeed have been practised in other programmes.

Methods:

The authors describe how they will assign the study group and the control group dependent on regions that the children live in. This does not seem to be a “randomised” method of sampling and I wonder if the authors could be clearer as to how this is randomised and exactly what they did in sampling for randomisation. Actually I can see later that it is discussed in a next section where the authors describe “doing a coin flip”.

We have used the term “quasi-randomized” study already in the title, indicating that this is not a study in which every child is randomized, but randomization is done by geography for reasons explained in

the manuscript. When the grouping has been carried out, we will “flip a coin” to decide which group will get the intervention the first year, and which group will be on the waiting list.

Recruitment:

Children from a previous study with CP are to be recruited and then some new children. How will a diagnosis of CP be confirmed in these new children and could the team mention how it was confirmed in the previous study?

We have used the definition of CP Surveillance in Europe for both cohorts. This has now been inserted in the text together with the proper reference.

Outcomes

There is a good range of outcome measures which are being utilised and it is great to see this list of measures. This will be incredibly helpful for other researchers. Should the team mention their primary outcome more clearly (as mentioned in the sample size calculation).

We have added a statement on the main outcome measure under the outcome measures.

Steering committee:

I wondered if the team would consider having a parent carer on the steering committee? This might add weight to the study.

This is a good suggestion for such an individual will help provide opportunities to promote the Akwenda programme in the community and engage other members of the communities in supporting its goals, provide advice, ensure delivery of the project outputs and the achievement of project outcomes. We will consult on this possibility to identify a parent carer, (not involved in the project) to confidently serve this purpose in the respective communities.

Inclusion of a parent carer in the Steering Committee has now been added.

Please could the authors go over the grammar – it may just be a matter of English but there are places where things are not reading quite right e.g. Page 3 “Strengths and Limitations”; “will be parent-led training workshops which account for the scarcity...” rather than “which accounts for” .

The authors, several of whom have English as their first language, have read through and checked the grammar.

VERSION 2 – REVIEW

REVIEWER	Stephanie Tremblay McGill University, Canada
REVIEW RETURNED	17-Feb-2021

GENERAL COMMENTS	Thank you for addressing the comments in your revision. Congratulations on submitting an interesting protocol and best of luck for the project!
---

REVIEWER	Melissa Gladstone University of Liverpool
REVIEW RETURNED	17-Feb-2021

GENERAL COMMENTS	The authors have responded to my comments and I have little extra to add. I do agree with reviewer 1, that some more clarity over the complexity of a programme such as this and a discussion on how this might possibly be implemented in other settings would be useful to add to the discussion. The programme is quite detailed and complex and may or may not be feasible or sustainable in other settings. I suppose a point in the discussion making this clear and that further research using implementation
---

	science would help this to understand what is actually scaleable in the future.
--	---